# PTEN modulates urinary tract infection susceptibility and shapes urothelial antibacterial defenses

Aaron Simoni[1,*], M Skye Bochter[1,*], Sarah Linn-Peirano[1,2], Kristin Salamon[1], Brian Becknell[1,3], Hanna Cortado[1], Ashley R Jackson[1,3], Laura Schwartz[1,3], John David Spencer[1,3]

Despite advancements in our understanding of urinary tract infection (UTI) pathogenesis, UTIs remain a leading cause of morbidity, partly because of an incomplete understanding of the molecular pathways governing bladder antibacterial defenses. Here, we demonstrate that phosphatase and tensin homolog (PTEN), a negative regulator of PI3K/Akt signaling, is a critical modulator of bladder urothelial immune defenses and vulnerability to UTIs caused by uropathogenic *Escherichia coli* (UPEC). PTEN silencing in human bladder urothelial cells increases susceptibility to UPEC in vitro, and urothelial-specific PTEN knockout mice exhibit increased bacterial titers in the urine, bladder, and kidneys after in vivo transurethral UPEC infection. Mechanistically, PTEN deficiency enhances Akt phosphorylation, amplifying NFκB and FAK activity. Silencing NFκB or FAK in PTEN-deficient cells restores resistance to UPEC. These findings establish PTEN as an important regulator of bladder urothelial defenses, balancing immune activation and urothelial structural integrity to protect against UTI.

## Introduction

Urinary tract infections (UTIs) are one of the most routine bacterial infections encountered in clinical practice. In their lifetimes, more than 50% of all females experience at least one UTI and 20–30% of females in this cohort will develop recurrent infections (1). Antibiotics are the mainstay of UTI treatment in children and adults. Repeated antibiotic exposure can lead to adverse long-term health outcomes, and antibiotic overuse accelerates the development of antibiotic resistance (2, 3, 4). Uropathogenic *Escherichia coli* (UPEC) is the most common pathogen causing UTI. Notably, *E. coli* infections account for half of the estimated global burden of antibiotic resistance and up to 90% of *E. coli* strains are resistant to at least one antibiotic (3). In the last 30 yr, UPEC have developed resistance to antibiotics routinely prescribed to treat UTI, reducing their efficacy and escalating the urgency to identify alternative therapeutic strategies (3, 4). This clinical reality highlights the importance of understanding antibacterial defenses that regulate UTI susceptibility.

The bladder urothelium provides sentinel defenses against uropathogens, functioning as a physical barrier that resists UPEC and an active participant in innate immunity (5, 6, 7). This multilayered epithelium secretes cytokines and chemokines that recruit immune cells to the site of infection and produces antimicrobial peptides that kill invading pathogens. In addition, pattern recognition receptors activate signaling pathways that orchestrate UTI defenses (6, 8, 9). The efficacy of these urothelial defenses can vary among individuals, influenced by genetic or molecular factors that are not fully understood (10). This variability highlights the importance of investigating molecular pathways that regulate urothelial responses to UPEC.

One signaling cascade recognized for its role in UTI defense is the phosphoinositide 3-kinase/Akt (PI3K/Akt) pathway (11, 12, 13). During an acute UTI, this pathway is activated when PI3K catalyzes the production of phosphatidylinositol (3,4,5)-triphosphate (PIP3), which subsequently activates the serine/threonine kinase Akt (14, 15). Phosphorylated Akt initiates downstream signaling events that regulate vital cellular processes, including metabolism, proliferation, and survival. Beyond these established roles, PI3K/Akt is a regulator of host defense (15, 16, 17, 18, 19, 20, 21). PI3K/Akt inhibition disrupts innate and inflammatory immune responses, increases infection risk, and reduces survival (17, 18, 22). In immune cells, PI3K/Akt hyperactivation enhances pathogen clearance, whereas its attenuation facilitates pathogen survival (23, 24, 25). Although these findings suggest that PI3K/Akt activation can be beneficial in host defense, other studies indicate that PI3K/Akt inhibition promotes pathogen clearance by mitigating immune dysregulation associated with sustained PI3K/Akt activation (26, 27). Furthermore, bacterial pathogens, including UPEC, can exploit 2PI3K/Akt signaling to invade host cells and establish intracellular niches, underscoring the dual-edged nature of this pathway (28, 29, 30, 31, 32, 33). Together, these findings suggest that precise PI3K/Akt regulation may be important for maintaining

---

[1]Kidney and Urinary Tract Center, The Abigail Wexner Research Institute at Nationwide Children's, Columbus, OH, USA  [2]College of Veterinary Medicine, University of Tennessee Institute of Agriculture, Knoxville, TN, USA  [3]Division of Nephrology and Hypertension, Department of Pediatrics, Nationwide Children's, Columbus, OH, USA

Correspondence: John.Spencer@nationwidechildrens.org
*Aaron Simoni and M Skye Bochter contributed equally to this work and are the co-first author.

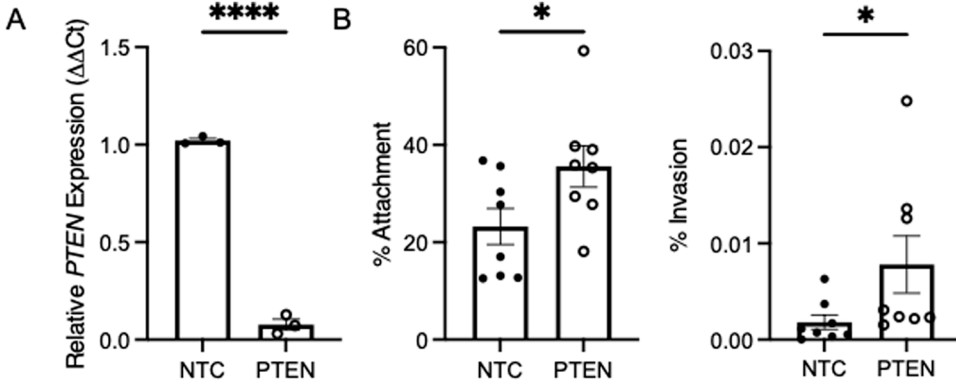

**Figure 1. PTEN silencing promotes UPEC infection in human urothelial cells.**
Human urothelial cells were transfected with a nontargeting control or PTEN siRNA pool. **(A)** qRT–PCR confirms PTEN silencing. Each point depicts mean *PTEN* expression from independent experiments performed in duplicate (*n* = 3/group). Graphs show the mean and SEM. **(B)** Transfected cells were challenged with UTI89. Shown are the percentage of bacteria adhering to the cellular surface (left) or invading the cells (right). Results are from eight independent experiments performed in duplicate. Graphs show the mean and SEM. Asterisks denote significant *P*-values for the pairwise comparison (*t* test). *P < 0.05 and ****P < 0.0001.

effective antibacterial defenses. Moreover, they highlight the complexity of PI3K/Akt responses, which may depend on the cell type involved and inciting pathogen (16).

Phosphatase and tensin homolog (PTEN), a dual-specificity phosphatase, serves as a negative PI3K/Akt regulator. Although much research has focused on the role of PTEN as a tumor suppressor, it also maintains immune homeostasis (34, 35, 36). Investigating how PTEN modulates PI3K/Akt signaling and its effects on urothelial antibacterial defenses could provide valuable insights into UTI pathogenesis. To explore this, we used both human urothelial cells and urothelial-specific PTEN knockout mice to define how PTEN influences UTI susceptibility and urothelial antibacterial defenses. Using these complementary and translational models, we demonstrate that urothelial PTEN regulates UPEC susceptibility by modulating PI3K/Akt activity and its downstream effectors, including NFκB and FAK.

## Results

### Silencing PTEN expression in human urothelial cells augments UPEC susceptibility

To investigate the impact of PTEN on urothelial antibacterial defenses, primary human urothelial cells were transfected with PTEN or control siRNA pools. qRT–PCR confirmed PTEN silencing 2 d after transfection (Fig 1A). 3 d after transfection, confluent cells were challenged with UPEC. PTEN silencing increased the percentage of UPEC adhering to and invading the cells (Fig 1B). Similarly, treating cells with bpV(HOpic), a pharmacologic PTEN inhibitor that suppresses its phosphatase activity, increased UPEC attachment and invasion while not impacting UPEC growth (Fig S1A–C) (37). These results indicate that PTEN loss or impaired activity enhances susceptibility of the urothelium to UPEC invasion.

### PTEN silencing alters antibacterial gene expression and enhances NFκB activity

To assess the impact of PTEN silencing on urothelial antibacterial responses, we profiled the expression of 84 antimicrobial response

genes using a RT$^2$ profiler PCR array in cells transfected with control or PTEN siRNA pools and challenged with UPEC. The array included genes encoding pattern recognition receptors, signaling nodes critical to host defense, cytokines, chemokines, and antimicrobial peptides. Among the genes profiled, 38 genes were down-regulated and 46 were up-regulated in PTEN-silenced cells (Fig 2A). Significantly down-regulated genes included *IL1B*, *IL6*, *CXCL8*, *MAP2K4*, *NAIP1*, and *TLR2*, whereas the significantly up-regulated genes included *AKT1*, *MAP2K3*, *PYCARD*, *RELA*, *SLC11A1*, and *SUGT1* (Table S1). STRING-based protein–protein interaction and signaling enrichment analyses of these significantly differentially expressed genes revealed a dysregulated network enriched for Akt- and NFκB-mediated effectors—including *RELA*, *IL1B*, *IL6*, and *CXCL8*—as central deregulated nodes in PTEN-deficient cells (Fig 2B) (38).

In support of this STRING interaction analysis, Western blot demonstrates that PTEN silencing increases urothelial Akt and NFκB P65 phosphorylation (Fig 2C). Furthermore, we quantified the expression of NFκB-mediated cytokines in conditioned media collected from control and PTEN-silenced cells before and after UPEC challenge. PTEN silencing resulted in suppressed concentrations of IL-1β, IL-6, IL-8, both before and after UPEC challenge, whereas TNFα concentrations were not significantly different (Fig 2D). These findings suggest that PTEN silencing enhances Akt and NFκB phosphorylation while suppressing the production of cytokines involved in urothelial antibacterial defense (39, 40, 41, 42).

### FAK activity increases with PTEN silencing

In addition to producing cytokines and chemokines to prevent UTI, the urothelium also forms an impermeable barrier maintained by a uroplakin plaque, tight junctions, focal adhesions, and cytoskeletal adaptors (7). To assess whether PTEN impacts these processes, we evaluated the expression of genes involved in barrier maintenance. qRT–PCR shows that PTEN silencing does not alter the expression of uroplakin or other tested barrier genes. Among the tested focal adhesion–related genes, we observed increased *PTK2* expression (Fig S2A and B). *PTK2* encodes FAK, a downstream effector of integrin receptors that regulates actin cytoskeletal remodeling and can be exploited by UPEC to promote

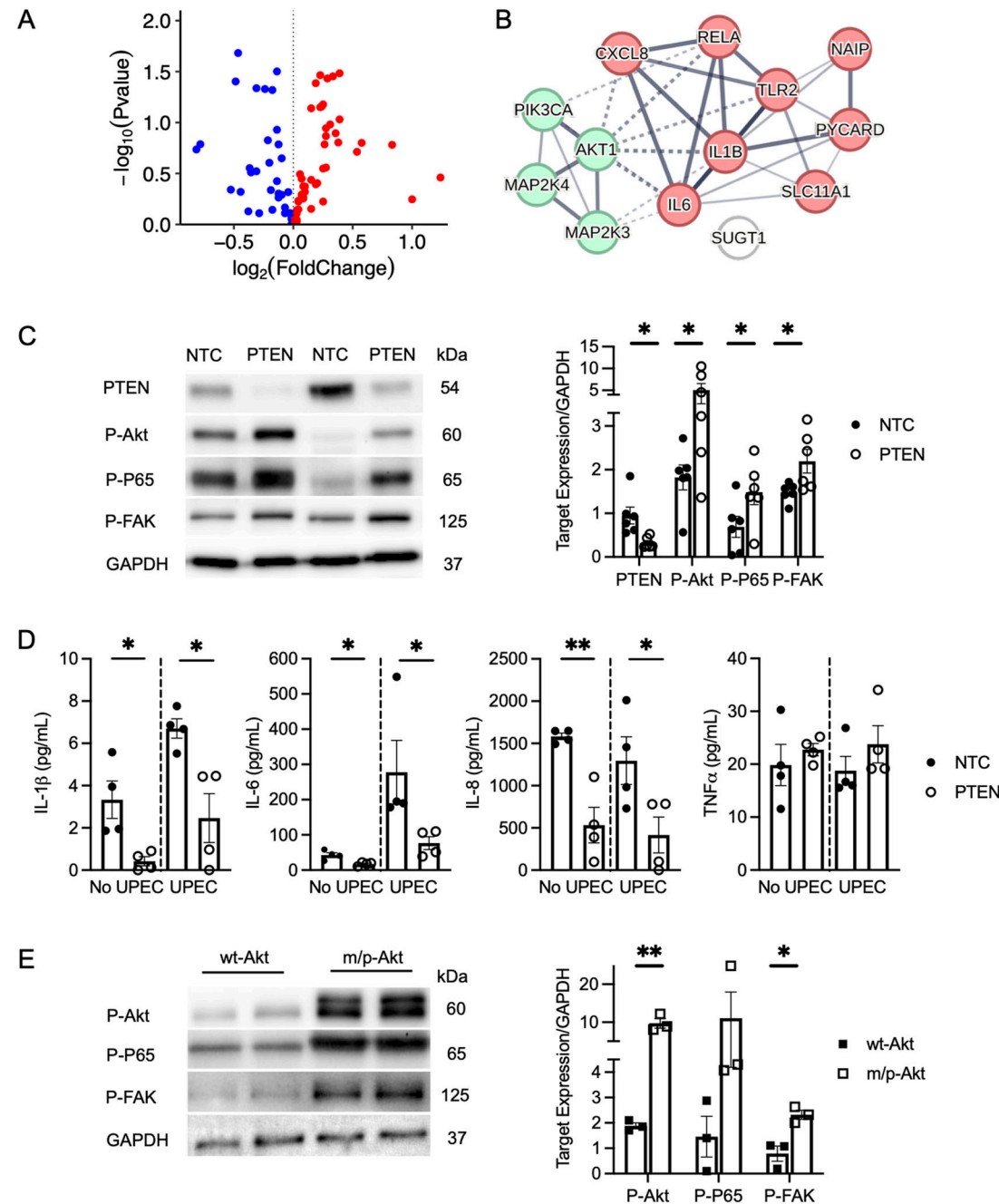

**Figure 2. PTEN silencing deregulates urothelial immune defenses and augments FAK expression.**
**(A, B)** Human urothelial cells were transfected with a nontargeting control or PTEN siRNA pool and challenged with UTI89 for 60 min. Antibacterial response gene expression was assessed using a RT² profiler PCR antibacterial response array (n = 3 samples/group). **(A)** Volcano plot illustrating differentially expressed genes in PTEN-silenced cells relative to controls. **(B)** Protein–protein interaction network was constructed using STRING based on the identified significantly differentially expressed genes (Table S1). Nodes represent gene products, with edges indicating functional (dashed lines) or physical (solid lines) interactions. Edge thickness corresponds to experimentally determined interaction confidence scores. Green nodes denote proteins with serine/threonine/tyrosine kinase activity, red nodes denote infectious disease-related proteins, and the open nodes indicate proteins that lack direct interactions or cluster membership. **(C)** Representative Western blot demonstrating PTEN, P-Akt (serine 473), P-P65 NFκB (serine 536), and P-FAK (tyrosine 397) expression in control and PTEN-silenced cells (left). Densitometry (right) shows the relative target abundance normalized to GAPDH. Results are from six independent experiments performed in duplicate. **(D)** Concentrations of IL-1β, IL-6, and IL-8 in conditioned media collected from control and PTEN-silenced cells before (no UPEC) and after bacterial (UPEC) challenge. Results are from four independent experiments performed in triplicate. Data are presented as the mean concentration and SEM. **(E)** Representative Western blot of P-Akt (serine 473), P-P65 NFκB (serine 536), P-FAK (tyrosine 397), and GAPDH in urothelial cells transfected with plasmids encoding wild-type (wt-Akt) or membrane-targeted Akt (m/p-Akt). Densitometry (right) shows relative target abundance normalized to GAPDH. Results are from three independent experiments performed in duplicate. **(C, D, E)** Asterisks denote significant P-values for the pairwise comparison (unpaired t test). *P < 0.05 and **P < 0.01.

urothelial invasion (13, 43, 44). Consistent with this, Western blot confirms that PTEN silencing increases FAK phosphorylation (Fig 2C). In addition, cytoskeletal adaptor genes *TLN1* and *TNS1* were significantly deregulated (Fig S2B). These findings suggest that PTEN loss activates FAK and triggers early cytoskeletal remodeling, potentially increasing vulnerability to UPEC.

### Sustained Akt activation promotes NFκB and FAK activity

To determine whether Akt activation alone drives NFκB and FAK activity, we transfected human urothelial cells with plasmids encoding either wild-type Akt (wt-Akt) or constitutively active, membrane-targeted Akt (m/p-AKT) (45, 46). Western blot demonstrates that m/p-AKT–transfected cells have elevated Akt phosphorylation and increased activity of both NFκB and FAK (Fig 2E), suggesting that PTEN silencing promotes Akt-dependent NFκB and FAK signaling pathways.

### Silencing NFκB and FAK reduces UPEC susceptibility in PTEN-deficient cells

To evaluate whether increased NFκB and FAK expression in PTEN-deficient cells enhances UPEC susceptibility, we transfected human urothelial cells with control and siRNA pools targeting PTEN, FAK, NFκB, PTEN plus FAK, or PTEN plus NFκB. qRT–PCR confirmed single or dual target knockdown without affecting the expression of the other tested genes across all siRNA combinations (Figs 3A and S3). Compared with control cells, silencing PTEN and NFκB individually augmented UPEC attachment and invasion, whereas FAK silencing reduced UPEC invasion without impacting bacterial attachment (Fig 3B).

Dual silencing of PTEN plus NFκB or PTEN plus FAK partly mitigated UPEC attachment compared with PTEN-deficient cells. Similarly, dual target knockdown reduced UPEC invasion compared with PTEN-deficient cells. PTEN plus FAK silencing suppressed UPEC invasion comparable to PTEN silencing alone, whereas PTEN plus NFκB silencing decreased invasion relative to PTEN-deficient cells (Fig 3B). These results indicate that NFκB and FAK activity in PTEN-deficient cells is required to enhance vulnerability to UPEC.

### Characterization of urothelial PTEN knockout mice

To investigate whether these in vitro findings extend to UTI outcomes in vivo, we generated conditional knockout mice with PTEN deletion in their superficial and intermediate urothelial cell layers. Mice homozygous for the floxed *Pten* gene ($Pten^{fl/fl}$) were bred with tamoxifen-inducible uroplakin 2 Cre/ERT2 mice (Upk2-iCre), resulting in UPK2-iCre $Pten^{fl/fl}$ knockout mice (*Pten*-ko) and homozygous floxed control littermates (*Pten*-flox) lacking the Cre transgene (Fig 4A) (47, 48). In bladder urothelium isolated from PTEN knockout mice, PCR on genomic DNA confirmed Cre-mediated recombination deleting exon 5 and qRT–PCR identified reduced *Pten* expression (Fig 4B and C). Although PTEN transcripts were still detected in knockout mice, this likely reflects *Pten* expression in basal urothelial cells where the Upk2-iCre is not expressed (49). To confirm, we performed RNA in situ hybridization using probes specific to exon 5 of the *Pten* gene. In control mice,

*Pten* expression was visualized in all urothelial cell layers. In PTEN knockout mice, *Pten* was absent in superficial and intermediate cells, but still detected in basal urothelial cells (Fig 4D). Knockout mice had normal phenotypes, bladder histopathology, and no evidence of deregulated bladder uroplakin or barrier gene expression (Fig S4A and B).

In support of our in vitro findings, Western blot identified increased bladder urothelial Akt phosphorylation (Fig 4E) and ELISA measured increased FAK and NFκB P65 phosphorylation in non-infected urothelium collected from female PTEN knockout mice (Fig 4F). qRT–PCR showed suppressed transcript expression of *Cxcl1*, *Cxcl2*, *Il1*, *Il6*, and *Tnf* in noninfected urothelium from PTEN knockout mice (Fig 4G). After transurethral UPEC infection, we similarly observed increased NFκB P65 phosphorylation and suppressed *Il1* and *Il6* transcript expression in PTEN knockout mouse bladders (Fig S5A and B). Because NFκB target genes can affect cell recruitment to the bladder during UTI, we profiled phagocyte cell populations in UPEC-infected bladders using flow cytometry. 24 h after infection, we did not observe significantly different neutrophil, monocyte, and macrophage profiles between control and PTEN knockout mouse bladders (Fig S6).

### Urothelial PTEN deletion increases UTI susceptibility

To evaluate whether PTEN deletion impacts UTI vulnerability in vivo, we transurethrally infected female PTEN knockout mice and control littermates with UPEC. 6 h after infection, PTEN knockout mice have a higher number of intracellular bacterial communities in the bladder (Fig 5A). At 16 and 48 h post-infection, knockout mice had greater UPEC titers in the urine and bladders compared with controls. By 48 h, the bacterial burden in the kidneys of PTEN knockout mice was also significantly greater (Fig 5B). Histopathologic analysis of bladder tissues 16 h after infection revealed significantly higher pathology scores. Infected PTEN knockout mouse bladders were more edematous and had increased inflammation and hemorrhage compared with controls—hallmarks of more severe acute infection (Figs 5C and D and S7A–F).

## Discussion

Key findings from these experiments establish PTEN as a critical regulator of urothelial antibacterial defenses against UPEC. Functionally, PTEN silencing in vitro increases UPEC attachment and invasion of human urothelial cells and PTEN deletion in suprabasal murine urothelial cells augments UPEC burden and bladder inflammation in vivo. Mechanistically, PTEN loss enhances Akt and NFκB while paradoxically suppressing the expression of proinflammatory cytokines involved in UTI defense (39). Deficient PTEN expression also augments FAK—an established mediator of UPEC invasion into the urothelium (Fig S8) (43, 51). Notably, silencing NFκB or FAK in PTEN-deficient cells rescues these effects, highlighting the impact of PTEN on immune regulation and urothelial integrity.

To our knowledge, the role of PTEN in UTI defense has not been investigated. However, studies in other systems suggest that PTEN

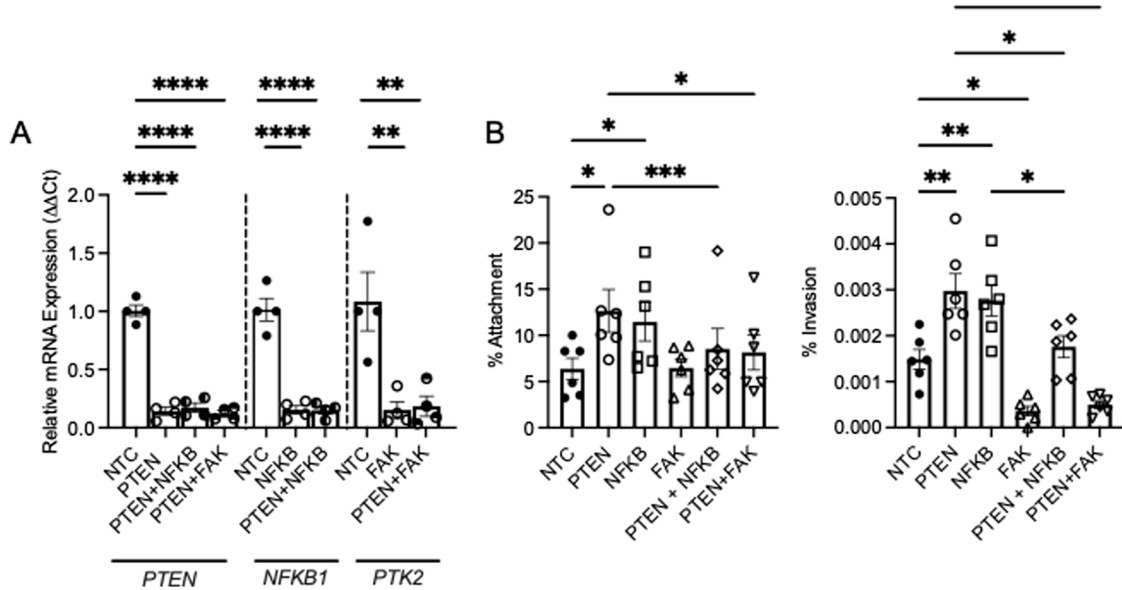

**Figure 3.  Silencing NFκB or FAK in PTEN-deficient cells reduces UPEC susceptibility.**
**(A)** Human urothelial cells were transfected with a nontargeting control, PTEN, NFκB (encoded by gene *NFKB1*), FAK (encoded by gene *PTK2*), PTEN plus NFκB, or PTEN plus FAK siRNA pools. qRT–PCR confirmed target silencing. Each point depicts mean target expression from independent experiments performed in duplicate (*n* = 4/group). Graphs show the mean and SEM. **(B)** Transfected cells were challenged with UTI89. Shown are the percentage of bacteria adhering to the cellular surface (left) or invading the cells (right). Results are from six independent experiments performed in duplicate. Graphs show the mean and SEM. Asterisks denote significant *P*-values for the pairwise comparisons (ANOVA). *$P$ < 0.05, **$P$ < 0.01, ***$P$ < 0.001, and ****$P$ < 0.0001.

modulates host susceptibility to infection. Mice with intestinal epithelial cell-specific PTEN deletion exhibit increased mortality, inflammation, proinflammatory cytokine production, intestinal mucosal injury, and disseminated infection after enteral *Salmonella typhimurium* challenge (52). In the genitourinary system, microscopy reveals that human PTEN null PC3 prostate cancer cells harbor increased intracellular *Mycoplasma* spp. after bacterial challenge. Similarly, UM-UC-3 bladder cancer cells lacking PTEN expression show greater susceptibility to *Mycobacterium bovis* bacillus Calmette–Guérin (BCG) invasion compared with PTEN-expressing MGHU4 bladder cancer cells. PTEN knockdown in MGHU4 cells via shRNA increased the fraction of intracellular BCG-infected cells, supporting a causal link between PTEN loss and enhanced epithelial permissiveness to intracellular pathogens (53).

Within immune cells, PTEN deletion or inhibition also has cell type–specific effects. PTEN-deficient lymphocytes have an overactive immune response and enhanced antibacterial immunity (36, 54). Pharmacologic PTEN inhibition in murine neutrophils increases reactive oxygen species production and bactericidal activity against *E. coli* (55). In contrast, PTEN-deficient murine macrophages exhibit reduced expression of proinflammatory cytokines, resulting in increased susceptibility to *Leishmania major* but improved killing of *Streptococcus pneumoniae* (56, 57). These findings highlight PTEN's broad regulatory role in pathogen susceptibility across different cell types (36, 52, 53, 54, 55, 56, 57, 58).

Several studies implicate PI3K/Akt in pathogen internalization and infection severity (16, 27, 33, 59, 60, 61, 62). In the context of UTI, PI3K/Akt impacts UPEC vulnerability. In vitro studies demonstrate

that PI3K/Akt activation is essential for UPEC invasion of 5,637 human bladder cancer cells or mpkIMCD mouse kidney cells (13, 14, 63). Specifically, UPEC activates host cell signaling networks, including PI3K/Akt, to promote bacterial internalization, whereas PI3K/Akt inhibition reduces UPEC invasion (13, 14). However, PI3K/Akt can also function as a defensive mechanism against UPEC. Our team previously showed that systemic PI3K/Akt inhibition heightens acute UPEC susceptibility in vivo (12). In addition, PI3K/Akt negatively regulates the expression of decay-accelerating factor (DAF), a complement regulatory protein that also functions as a receptor for *E. coli* expressing Dr fimbriae (Dr$^+$) (64, 65). In HeLa and Ishikawa epithelial cells, PI3K/Akt inhibition increases DAF expression and enhances bacterial adhesion, whereas PTEN inhibition activates PI3K/Akt, leading to DAF down-regulation and decreased Dr+ *E. coli* adhesion (32). These results underscore the dual role of PI3K/Akt in modulating UPEC–host interactions.

Beyond PI3K/Akt, PTEN also functions as a protein phosphatase that directly dephosphorylates FAK at tyrosine 397, thereby inhibiting its activity. Consistent with prior studies, we found that PTEN deficiency promotes FAK activation. Our findings also suggest that Akt hyperactivity further drives FAK expression (36, 66, 67, 68). FAK is a key component of the focal adhesion complex, coordinating signals from integrin-mediated cell–extracellular matrix interactions and receptor tyrosine kinases to regulate cell survival, adhesion, and cytoskeletal remodeling (66, 69). Here, we show PTEN silencing in human urothelial cells or deletion in murine apical urothelial cells activates FAK. Silencing FAK in PTEN-deficient cells reduces UPEC invasion, highlighting its central role in mediating UPEC invasion. Published studies have similarly

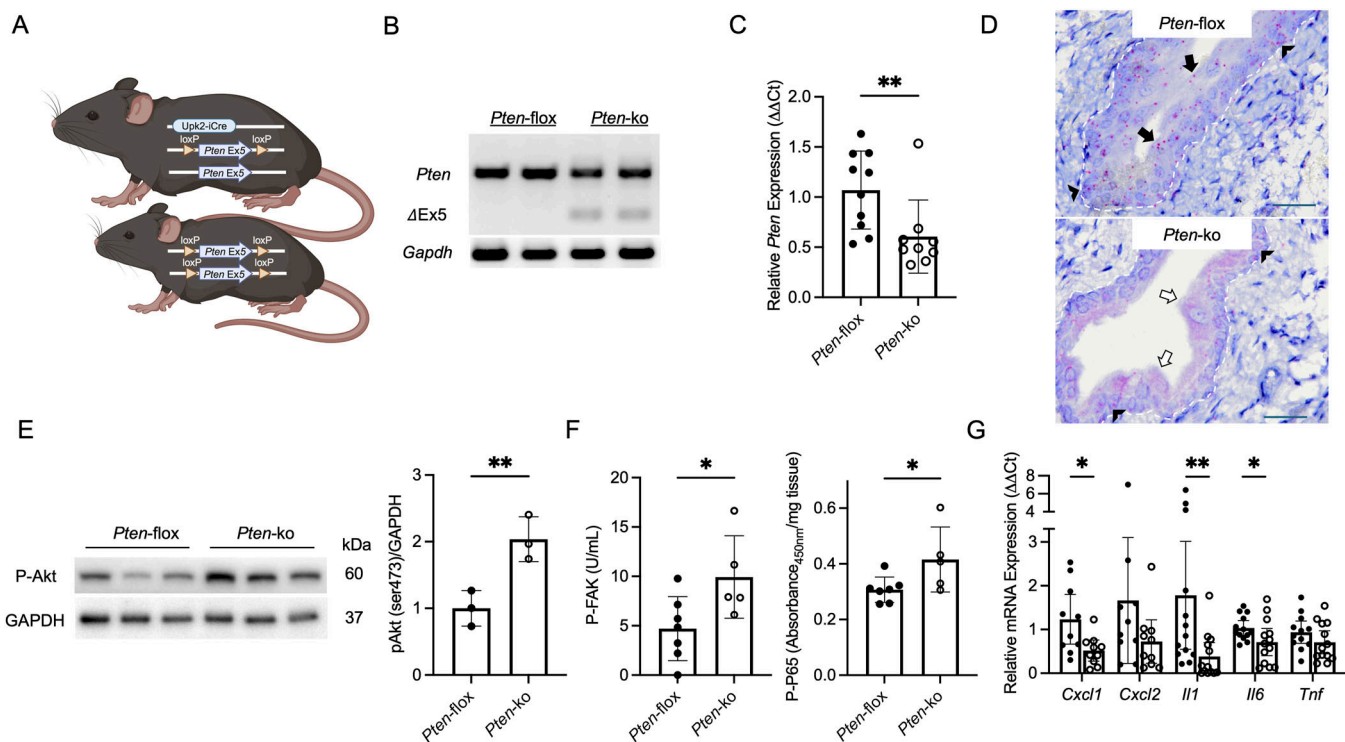

**Figure 4. PTEN deletion in superficial and intermediate murine urothelial cells.**
**(A)** Schematic illustrating the breeding strategy for *Pten* deletion using uroplakin 2 (Upk2)-iCre transgenic mice. **(B)** Representative conventional PCR shows a 464 base pair product (top band), indicative of the *Pten* floxed allele in mouse bladders, and a smaller 225 base pair product (middle band) indicative of Cre recombination and exon 5 deletion. The retained presence of the 464 base pair product in PTEN knockout mice is attributed to other bladder cell types not targeted by Cre recombinase. *Gapdh* is a loading control (bottom band). Each lane shows results from a separate mouse. **(C)** Relative *Pten* transcript expression in urothelium of control and knockout mice (*n* = 9–10 mice/genotype). Asterisks mark significance between genotypes (Mann–Whitney). **(D)** Representative in situ hybridization images show *Pten* expression (pink) in apical (black arrow) and basal urothelial cells (chevron) in control mice (top). In knockout mice (bottom), *Pten* is detected in basal urothelial cells (chevron) but is absent in apical urothelial cells (open arrow). Scale bar = 20 $\mu$m. 600X magnification. The dashed line marks the urothelial and stromal interface. In situ hybridization was completed on four bladders/genotype. **(E)** Representative Western blots probed for P-Akt (Ser473) and GAPDH using noninfected urothelium from control and PTEN knockout mice (left). Each lane depicts protein expression from a separate mouse. Densitometry (right) showing relative target abundance normalized to GAPDH. Asterisks denote significant *P*-values for the pairwise comparison (unpaired *t* test). **(F)** Expression of (left) P-FAK (tyrosine 397) and (right) P-P65 NFκB (serine 536), measured by ELISA, in noninfected bladder urothelium collected from control and PTEN knockout mice. Each point depicts mean target expression in a unique mouse (*n* = 5–7 mice/genotype). Graphs show the mean expression and SEM. **(G)** Cytokine mRNA expression in noninfected bladder urothelium of control (closed circles) and PTEN knockout mice (open circles, *n* = 13 mice/genotype). **(F, G)** Asterisks show significance between genotypes (Mann–Whitney). *P < 0.05 and **P < 0.01.

demonstrated that FAK is essential for urothelial invasion by UPEC. In 5,637 cells, siRNA-mediated FAK knockdown or pharmacologic inhibition with PF573228 or FAK inhibitor 14 reduces UPEC invasion (43, 51, 70). *E. coli* invasion is also reduced in FAK-null mouse embryonic fibroblasts and human brain microvascular endothelial cells with deficient FAK signaling (43, 71). Beyond *E. coli*, FAK also modulates the invasion of facultative intracellular pathogens, including Salmonella, Shigella, and Yersinia (51, 72, 73, 74, 75).

During UPEC infection, FAK is activated downstream of α3- and β1-integrin engagement, which occurs when the type I pilus adhesion FimH binds to the urothelial surface (13, 43). This integrin–FAK interaction initiates a signaling cascade involving different adaptor proteins and kinases that orchestrate actin cytoskeletal remodeling. Furthermore, FAK activation reinforces PI3K signaling by recruiting PI3K to focal adhesion complexes, amplifying downstream remodeling signals (Fig S8) (13, 43, 50, 76, 77, 78). These rearrangements enable the host cell to wrap around and internalize adherent bacteria through a zippering mechanism. In parallel, FAK signaling coordinates the recruitment of vesicular

membranes, including endosomes and Rab27b-positive compartments, to supply added membrane for the zippering process (50, 79). Through this dual role in cytoskeletal remodeling and membrane trafficking, FAK serves as a key convergence point for the host invasion machinery exploited by UPEC. Although the requirement for FimH in the PTEN-deficient context was not directly tested, our data suggest that PTEN-deficient cells exhibit a FAK-dependent invasive phenotype that is contingent on type 1 fimbria engagement. Future studies using FimH-deficient UPEC strains in PTEN-deficient cells may be valuable to determine whether FimH is required for the enhanced invasion phenotype.

Given its central role in UPEC invasion, FAK inhibition has been explored as a therapeutic strategy to reduce UPEC susceptibility. *Lactuca indica* extract from Vietnamese dandelion interferes with bacterial adherence and FAK activation, reducing UPEC invasion in vitro (70). In addition, the plant-derived phenolics caffeic acid phenethyl ester (CAPE) and resveratrol inhibit FAK and reduce UPEC invasion of human 5,637 bladder cells (51). Intravesical resveratrol administration with UPEC infection reduces intracellular

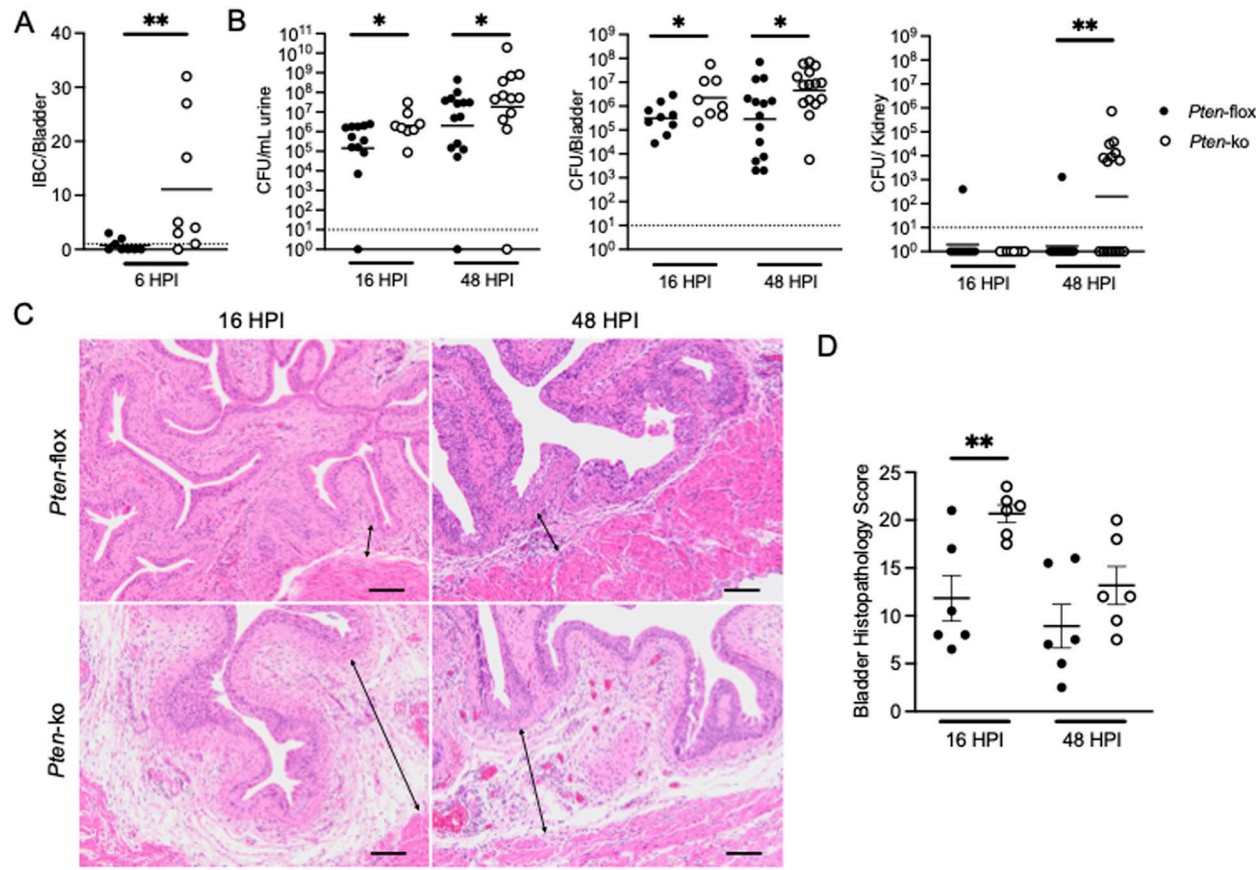

**Figure 5.  PTEN deletion in bladder urothelium increases urinary tract infection (UTI) susceptibility and exacerbates tissue injury.**
**(A, B)** Female PTEN knockout mice and littermate controls were transurethrally infected with UTI89. **(A)** Bladder intracellular bacterial communities enumerated 6 h after infection. Each point denotes the number of intracellular bacterial communities detected in a unique mouse bladder (n = 8 mice/genotype). The horizontal line shows the mean of each group. The dotted line represents the limits of detection. **(B)** Urine (left), bladder (center), and kidney (right) UTI89 burden 16 and 48 h post-infection (HPI). The horizontal line shows the geometric mean of each group. Each point denotes UPEC burden in a unique mouse. The dotted line represents the limits of detection. **(C)** Representative micrographs of bladder tissue from UPEC-infected control and PTEN knockout mice. Bladders from PTEN knockout mice demonstrated increased inflammation and hemorrhage within the lamina propria along with significantly increased edema compared with controls (black double arrows). Scale bar = 100 μm. 100x magnification. **(D)** Bladder histopathology scores in UPEC-infected mice as calculated from Table S3. Graphs show the mean score and SEM. Each point denotes a score in a unique mouse (n = 6 bladders/genotype and 4–5 kidneys/genotype). Asterisks identify significant P-values for the indicated pairwise comparisons (Mann–Whitney). *P < 0.05 and **P < 0.01.

UPEC titers recovered from mouse bladders ([51]). Although our data show that silencing FAK in PTEN-deficient cells reduces UPEC invasion in primary human urothelial cells, we have not tested pharmacologic FAK inhibition rescues the PTEN-deficient invasion phenotype in vivo. These experiments will require the access to clinically safe and bio-available FAK inhibitors, as well as optimization of drug delivery, dosing and administration, timing of administration, tissue penetration, and pharmacodynamic profiling—parameters that are not well established as this class of compounds continues to be developed for clinical use ([80], [81]). If optimized, FAK inhibition could help prevent UPEC from invading and persisting within the bladder or kidney, rendering it more susceptible to host immune defenses and antibiotics that poorly access intracellular niches.

Our data also indicate that PTEN deletion and Akt over-expression activate NFκB. Reducing NFκB expression in PTEN-deficient cells decreases UPEC vulnerability. The urothelium responds to UPEC by activating NFκB, which initiates immune responses to clear infection ([82], [83]). Recently, we demonstrated

that impaired NFκB signaling disrupts urothelial barrier genes and suppresses the expression of antimicrobial peptides and cytokines involved in UTI defense ([49]). In addition, we found that Akt regulates NFκB partly through degradation of the NFκB inhibitor, IκB ([49], [84], [85]). In the kidney, NFκB impairment enhances UPEC attachment and invasion while suppressing antimicrobial peptide expression ([86]). Conversely, sustained NFκB activation can drive maladaptive inflammatory responses and heighten UTI susceptibility ([82]). These findings, along with our current data, underscore the importance of maintaining a balanced immune response that can be engaged to eradicate pathogens.

We further demonstrate that PTEN deficiency suppresses cytokines critical for UTI defense, including IL-1β, IL-6, and IL-8. Similar cytokine suppression has been observed in PTEN deletion in myeloid cells ([56], [57], [58]). In the context of UTI, precise IL-1β regulation is essential as IL-1β overactivation promotes cystitis ([40]). Although several studies suggest that IL-1β promotes a protective and beneficial inflammatory response that controls UPEC

colonization, others suggest that mice lacking IL-1$\beta$ are protected from UTI (87, 88, 89, 90). IL-6 is another key regulator of UTI defense as mice lacking IL-6 exhibit increased susceptibility to UPEC (39). Similarly, murine IL-8 receptor homolog knockout mice fail to clear UPEC from their bladders and kidneys, leading to bacteremia (42, 91). Clinical data implicate a protective role of IL-8 and its receptor CXCR1 in UTI. Children prone to pyelonephritis and recurrent UTI exhibit lower expression of CXCR1 compared with age-matched controls (42). Moreover, polymorphisms in *IL8* and its receptor *CXCR1* are associated with an increased risk of pyelonephritis (92, 93, 94). Together, these findings suggest that PTEN deficiency compromises UTI defense by suppressing cytokine-mediated immune responses.

We were surprised to observe suppressed IL-1$\beta$, IL-6, and IL-8 expression despite detecting increased Akt and NF$\kappa$B activity, which can enhance cytokine production (10, 82, 95). However, sustained Akt or NF$\kappa$B could ultimately reduce cytokine expression through negative feedback mechanisms, including changes in transcription factor activity, epigenetic regulation, and metabolic reprogramming that dampen inflammatory signaling (96, 97, 98). Alternatively, the suppressed cytokine expression observed in our study could be driven by PTEN-mediated pathways independent of Akt or NF$\kappa$B, involving effectors we did not interrogate.

We acknowledge this study has limitations. First, we employed a targeted approach by focusing on specific signaling pathways regulated by PTEN. Although this strategy enabled us to define PTEN-mediated mechanisms affecting UTI susceptibility, a broader transcriptomic, metabolomic, or proteomic approach could reveal additional pathways and downstream effectors involved in host defense. Second, although we demonstrate that PTEN loss promotes UPEC susceptibility in human and mouse models, the clinical impact remains uncertain. Validation in human bladder tissue from patients prone to UTI could strengthen the relevance of our findings. In addition, our study evaluated short-term UTI outcomes, leaving the role of PTEN in recurrent or chronic UTI unexplored. Because PTEN regulates both inflammatory and extracellular matrix functions, its contribution to persistent UPEC reservoirs or recurrent UTI pathogenesis warrants further investigation. Finally, although our findings suggest that PTEN regulates signaling pathways implicated in UTI, strategies targeting PTEN, Akt, or FAK for therapeutic intervention require careful consideration. Given the essential roles of these pathways in cellular homeostasis, broad inhibition could have unintended effects.

To conclude, this study identifies PTEN as a regulator of bladder antibacterial defenses. Mechanistically, PTEN silencing or deletion increases UPEC susceptibility by activating Akt, NF$\kappa$B, and FAK signaling while suppressing proinflammatory cytokine expression. Collectively, these findings provide new insights into host–pathogen interactions and offer new insights into potential therapeutic targets for UTI.

# Materials and Methods

## Reagents and tools

Details of the reagents and mice used can be found in Table S2.

## Experimental models

### Human urothelial culture

Primary human bladder urothelial cells, obtained from an 80-yr-old man (HBLAK cells, CELLnTEC Advanced Cell Systems), were cultured in CnT-Prime Epithelial Proliferation media (CELLnTEC Advanced Cell Systems) at 37°C in the presence of 5% $CO_2$ (99).

### Mice

Mouse experiments were performed in accordance with Institutional Animal Care and Use Committee rules and regulations. All mice were maintained on standard rodent chow ad libitum with free access to water under controlled temperature and humidity with 12-h light-and-dark cycles. PTEN knockout mice were generated by crossing C57BL/6J mice homozygous for the floxed *Pten* exon 5 (Jackson Laboratory) with tamoxifen-inducible Upk2-iCre C57BL/6J mice (Jackson Laboratory) (47, 100). Cre(+);*Pten*$^{fl/+}$ progeny were bred with *Pten*$^{fl/fl}$ mice, generating Cre(+);*Pten*$^{fl/fl}$ and Cre(−);*Pten*$^{fl/fl}$ offspring. At 6 wk of age, Cre-positive and Cre-negative offspring received three doses of intraperitoneal tamoxifen dissolved in corn oil (100 mg/kg; Cayman Chemical). After an 11-d tamoxifen washout period, mice were subjected to downstream studies. Mice were genotyped according to published protocols (47, 100).

## Method details

### In vitro gene silencing

siRNA pools targeting *PTEN*, *NFKB1*, *PTK2*, and a nontargeting control were purchased (ON-TARGETplus SMARTpool; Dharmacon). Human urothelial cells were cultured to 95% confluency and transfected with a mixture of siRNA, DharmaFECT transfection reagent (Dharmacon), and culture media per the manufacturer's recommendations. 48 h after transfection, cellular RNA was isolated for qRT–PCR. 72 h after transfection, cells were used for protein assays or UPEC infection experiments.

### In vitro Akt overexpression

Plasmids encoding pCMV5-hemagglutinin (HA)-PKB$\alpha$ (wt-Akt) or pCMV5-HA-membrane–targeted PKB$\alpha$ (m/p-AKT) were purchased from D. Alessi (University of Dundee, Dundee, United Kingdom) and transfected into primary human urothelial cells using Lipofectamine LTX according to the manufacturer's instructions (Invitrogen) (45, 46). 48 h post-transfection, culture media were collected for ELISA analysis and protein lysates were collected for Western blot.

### In vitro UPEC infection assays

Human urothelial cells were cultured in 24-well plates and either transfected with siRNA or treated with the indicated compounds before UPEC challenge. Cells were infected with 10 multiplicity of infection UPEC (strain UTI89). UTI89 is type I–piliated UPEC strain isolated from a patient with cystitis (101). UPEC attachment and invasion assays were performed as described previously (49, 102, 103). To assess the potential effects of bpV(HOpic) on bacterial growth, UPEC were added to complete culture media in 96-well plates with vehicle control (water), increasing concentrations of

https://doi.org/10.26508/lsa.202503292   vol 8 | no 10 | e202503292

bpV(HOpic) or gentamicin (100 $\mu$g/ml). After 6-h incubation, bacterial turbidity was measured at 600 nm ($OD_{600}$) using a Synergy HT multimode microplate reader (BioTek Instruments) (104).

### Mouse UTI model

Female mice aged 6–8 wk were infected by transurethral catheterization with $10^7$ CFU UTI89 as previously published (12, 49). At the indicated time points after inoculation, mice were anesthetized and euthanized via cervical dislocation. Intracellular bacterial communities were enumerated as previously described (39, 105). To determine bacterial burden, urine was collected, the bladder and kidneys were aseptically harvested, and UPEC colonies were enumerated by plating serial dilutions. Counts less than 9 CFU at any dilution were below the limits of quantification.

### Mouse bladder histopathology and immunostaining

Bladders were fixed in 4% PFA, then fixed in 70% ethanol, and then processed for paraffin sectioning. Slides were sectioned at 5 $\mu$m. Sections used for histopathology were stained with hematoxylin and eosin. Four representative sections were evaluated per organ, and bladder pathology scoring was modified from a previous scoring system (Table S3 and Fig S7) (106).

Images were acquired using an Olympus BX53 microscope equipped with a 10× ocular objective and various objective lenses. Total magnification is reported as the product of ocular and objective magnification (e.g., 10X ocular with 40X objective yields 400X total magnification). Objectives used for imaging included 10X/0.30 numerical aperture (NA), 20X/0.5 NA, 40X/0.65 NA, and 60X/0.90 NA dry lenses, as well as a 100X/1.25 NA oil immersion lens.

### In situ hybridization using mouse bladder

*Pten* was detected in mouse bladder sections using RNAscope Chromogenic Assay following the manufacturer's protocol (Advanced Cell Diagnostics). Deparaffinized and rehydrated slides were pretreated with the RNAscope Hydrogen Peroxide reagent for 10 min, RNAscope Target Retrieval buffer for 20 min, and Protease Plus reagent for 30 min. Samples were hybridized with the Mm-*Pten*-E5 probe targeting exon 5 or *dapB* probe (negative control) for 2 h (Advanced Cell Diagnostics). The RNAscope 2.5 High Definition-Red Assay was used to detect probe-target hybridization. Slides were counterstained in 50% Gill's Hematoxylin I (Thermo Fisher Scientific). Coverslips were applied with VectaMount (Vector Laboratories, Inc.). Images were captured using a Nikon Ti2-E microscope and DS-RI2 camera (Nikon Instruments Inc.).

### RNA isolation, real-time PCR, and antimicrobial response arrays

RNA was isolated using RNeasy Plus Mini Kit (QIAGEN) from human cell lysates or mouse tissue following the manufacturer's instructions. For qRT–PCR, cDNA was generated using the Verso cDNA synthesis kit (Thermo Fisher Scientific). qRT–PCRs were performed with 7,500 Real-Time PCR System (Applied Biosystems) as described previously (104). Each reaction included cDNA, Absolute Blue qPCR SYBR Mix (Thermo Fisher Scientific), and target-specific primers (Tables S4 and S5). For the antibacterial response arrays, cDNA was generated using RT2 First Strand Kit (QIAGEN). Human Antibacterial Response Arrays (QIAGEN) were performed following the manufacturer's instructions and analyzed using QIAGEN's GeneGlobe Data Analysis Center. Samples were normalized to the average geometric mean of the housekeeping genes.

### Western blot and ELISA

Western blot was performed as previously described (86, 103). Antibodies directed against the following targets were used: PTEN (Cell Signaling), P-Akt (Ser473) (Cell Signaling), NF$\kappa$B P-p65 (Ser536) (Cell Signaling), P-FAK (Tyr397) (Cell Signaling), and GAPDH (Cell Signaling). Western blots were imaged on ChemiDoc MP (Bio-Rad Laboratories). Densitometry was performed using ImageLab software, and adjusted volume band intensity values for target proteins were normalized to corresponding adjusted volume band intensity values for GAPDH (86). Commercial ELISA assays quantified concentrations of P-FAK (Tyr397) (Invitrogen), P-NF$\kappa$B P65 (Ser536; Cell Signaling), IL-1$\beta$, IL-6, IL-8 (R&D Systems) following the manufacturer's instructions.

### Flow cytometry

Bladders were harvested 24 h after transurethral UTI. Bladder tissues were minced and dissociated in DMEM/F-12 supplemented with 100 mg/ml collagenase I and 10 mM Hepes. After enzymatic dissociation, single-cell suspensions were incubated with an anti-mouse Fc receptor antibody to block nonspecific antibody binding. Cells were then stained with blue-fluorescent reactive dye (L23105; Life Technologies) for 20 min to remove dead cells. Cells were then stained for cell surface receptors in FACS buffer for 15 min at 4°C with fluorescent monoclonal antibody combinations (Table S2). Stained cells were collected on a LSR II cytofluorometer (BD) and data analyzed using FlowJo software as previously published (TreeStar) (49). Absolute cell numbers were calculated using CountBright absolute counting beads (Invitrogen).

### Statistics

The D'Agostino–Pearson Omnibus or Shapiro–Wilk tests were used to assess normality, defined as a $P > 0.05$. All in vitro data exhibited a normal distribution, whereas in vivo data did not. As indicated in the figure legends, an unpaired $t$ test was used for two-way comparisons, and ANOVA was applied for multiple comparisons of normally distributed data. Otherwise, the nonparametric Mann–Whitney $U$ test was used. Differences between groups with a $P < 0.05$ were regarded as statistically significant.

# Data Availability

All data supporting the findings of this study are available within the article and its supplementary information files. Source data are available from the corresponding author upon reasonable request.

# Supplementary Information

# Acknowledgements

We thank Birong (Rollin) Li for assistance with the mouse infections and Drs. Hancong (Chloe) Tang and Steve Rust for guidance with statistical analysis. This work was supported by the National Institutes of Health (NIDDK) R01 DK114035 and R01 DK115737 (to JD Spencer).

## Author Contributions

A Simoni: conceptualization, data curation, formal analysis, investigation, methodology, project administration, and writing—original draft, review, and editing.
MS Bochter: conceptualization, data curation, formal analysis, supervision, validation, investigation, visualization, methodology, project administration, and writing—original draft, review, and editing.
S Linn-Peirano: data curation, formal analysis, methodology, and writing—original draft, review, and editing.
K Salamon: data curation, formal analysis, investigation, methodology, and writing—original draft, review, and editing.
B Becknell: data curation, formal analysis, and writing—original draft.
H Cortado: data curation, formal analysis, methodology, and writing—original draft.
AR Jackson: data curation, formal analysis, supervision, investigation, and writing—original draft, review, and editing.
L Schwartz: data curation, formal analysis, supervision, visualization, methodology, project administration, and writing—original draft, review, and editing.
JD Spencer: conceptualization, data curation, formal analysis, supervision, funding acquisition, validation, investigation, visualization, methodology, project administration, and writing—original draft, review, and editing.

## Conflict of Interest Statement

The authors declare that they have no conflict of interest.

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
