## [Reviewer comments · Life Science Alliance]

Life Science Alliance

PTEN modulates urinary tract infection susceptibility and shapes urothelial antibacterial defenses

Aaron Simoni, M. Skye Bochter, Sarah Linn-Peirano, Kristin Salamon, Brian Becknell, Hanna Cortado, Ashley Jackson, Laura Schwartz, and John David Spencer

DOI: <https://doi.org/10.26508/lsa.202503292>

Corresponding author(s): *John David Spencer, Nationwide Children's Hospital*

Review Timeline:

Submission Date:	2025-03-04
Editorial Decision:	2025-04-25
Revision Received:	2025-06-08
Editorial Decision:	2025-07-04
Revision Received:	2025-07-11
Accepted:	2025-07-16

Scientific Editor: Sarita Hebbar

Transaction Report:

April 25, 2025

Re: Life Science Alliance manuscript #LSA-2025-03292-T

Dr. John David Spencer
The Research Institute at Nationwide Children's Hospital
Pediatrics
700 Children's Drive
Columbus, OH 43016

Dear Dr. Spencer,

Thank you for submitting your manuscript entitled "PTEN modulates urinary tract infection susceptibility and shapes urothelial antibacterial defenses" to Life Science Alliance. The manuscript was assessed by three expert reviewers, whose comments are appended to this letter.

All the reviewers find this work of potential value to the community. We invite you to submit a revised manuscript addressing the reviewers' comments.

When submitting the revision, please include a letter addressing all the reviewers' comments point by point. While a rebuttal must respond to all points in some form, additional data to resolve these points is not required.

Thank you for this interesting contribution to Life Science Alliance. We are looking forward to receiving your revised manuscript.

Sincerely,

Sarita Hebbar, PhD
Scientific Editor
Life Science Alliance
<http://www.lsjournal.org>

-- By submitting a revision, you attest that you are aware of our payment policies found here: <https://www.life-science->

B. MANUSCRIPT ORGANIZATION AND FORMATTING:

Reviewer #1 (Comments to the Authors (Required)):

Title: PTEN modulates urinary tract infection susceptibility and shapes urothelial antibacterial defenses

This manuscript examines the role of PTEN in uropathogenic *E. coli* (UPEC) UTI susceptibility. It has a well-done and logically flowing blend of in vitro and in vivo experiments that establish the importance of PTEN in urothelial cells. Some data are presented to illustrate the mechanisms by which PTEN could be affecting UPEC invasion of urothelial cells and UTI. Each main point of the paper is strongly supported by the data. Appropriate statistical tests are performed. The manuscript is exceptionally well written and the figures are clearly presented. Supplementary figures and tables are also provided to support the main figures. This study will be of substantial interest to investigators in the UTI field and likely also to others interested in urothelial biology.

I have only a few minor comments and suggestions:

Page 5: "These results indicate that PTEN loss or impaired activity enhances UPEC susceptibility."

I suggest changing to "enhances susceptibility of the urothelium to UPEC invasion" for clarity.

Page 5: "STRING protein-protein interaction and signaling enrichment analyses of these significantly differentially expressed genes revealed a dysregulated network with Akt and NFkB as central activated nodes in PTEN deficient cells."

I do not see NFkB as one of the nodes in the figure, as is suggested should be true by the text. Please specify which nodes implicate NFkB and why.

Page 7: I see errors in the description of Fig 3B. For example, it is stated that "PTEN and FAK silencing suppressed UPEC invasion comparable to FAK silencing alone" but this comparison is not indicated by a bar or star(s) on the graph. I think the authors likely meant to say comparable to PTEN silencing alone. Also it is stated that PTEN and NFkB silencing "remained elevated compared to controls" but the PTEN + NFkB and NTC bars are roughly the same height and no significant difference is indicated on the graph. Please make the necessary corrections to the graph and/or text descriptions.

Page 9: "Deficient PTEN expression also augments FAK - an established mediator of UPEC invasion into the urothelium."

It would be helpful for the authors to provide a few more sentences describing prior studies that established FAK as a mediator of UPEC invasion. What experiments were done, etc. This will allow the reader to place this study into the broader context of the field.

Page 9: "PTEN deletion enhances Salmonella typhimurium susceptibility." And "PTEN loss exhibit increased susceptibility to Mycoplasma spp. and Mycobacterium bovis Bacillus Calmette-Guerin infection."

Were these intracellular infections, like with UPEC in the present study? Again, a few more details of the experiments would be helpful.

Page 10: "The plant-derived phenolics caffeic acid phenethyl ester (CAPE) and resveratrol inhibit FAK and reduce UPEC invasion of human bladder epithelial cells.50 Additionally, Lactuca indica extract from Vietnamese dandelion interferes with bacterial adherence and FAK activation, thereby reducing UPEC invasion."

Since the authors have mentioned these compounds, it begs the question whether they would work to reduce UPEC invasion in vivo. While I do not think it is a requirement for this manuscript to have additional data, it would certainly be nice if a FAK inhibitor reversed the UTI severity phenotype in the PTEN conditional knockout mice. I think many readers will have this same thought. I encourage the authors comment on whether they have attempted this in their mice, why they have not chosen to, or whether they plan to in future studies.

Finally, kudos to the authors (or the journal if it was in the author instructions?) for providing the helpful information in the supplemental figure legends indicating which main figures they connect with. This is very helpful, and I wish all supplemental figure legends required this.

Reviewer #2 (Comments to the Authors (Required)):

The authors present a well written and concise study of mechanism by which PTEN and its downstream targets impact susceptibility to UTIs caused by UPEC. This study builds on their previous work dissecting this pathway and demonstrates how PTEN works upstream to regulate PI3K/Akt, and therefore its downstream effectors NFkB and FAK, to modulate UTI susceptibility. This work represents an important step towards understanding the host-pathogen mechanisms that contribute to host susceptibility to UPEC UTI. While there are many factors that have been identified that contribute to host susceptibility, this study carefully indicates a role for the PTEN pathway in modulating the inflammatory response to appropriately respond to UPEC and help clear the infection, which may lead to therapies that can help to finely tune the immune system to reduce recurrence and protect susceptible hosts. The authors detail critical experiments to support their conclusions; however, a couple key issues should be addressed to strengthen the manuscript.

Major Issues/Key Experiments:

1. Additional details for figure 2 about controls used would help clarify the results for the reader. Specifically, for the inflammatory genes differentially regulated (2A and B) between NTC and PTEN UPEC infected cells, are any of these genes differentially regulated in the uninfected cells prior to infection.
 - a. Small comment, but 2B only displays NFkB-mediated cytokines not NFkB itself. The text could be clarified in the sentence prior to 2B denoted in the manuscript (page 5) and/or that designation should be added to the figure legend to clarify which are NFkB related genes.
2. It would strengthen the conclusions that the pathway is linear from PTEN to NFkB/FAK if the authors included the relative transcript of PTEN during the silencing of downstream targets, confirming no feedback/crosstalk (figure 3). Same for NFkB and PTK2 silencing (page 6)
3. The authors discuss the implications for altered IL-8 and its receptor CXCR1 in human UTI and specifically, the link to progression to pyelonephritis. Additionally, their human cell experiments show IL-8 is altered, while their mouse experiments indicate CXCI-1 is altered. While UPEC dissemination beyond the urinary tract in these models is rare (<5%), assessing dissemination in PTEN KO mice (spleen/heart/liver) could provide additional insight into mechanisms essential for preventing the spread from the bladder, which is arguably the most serious complication of UTI.

Minor:

Results:

Page 8, Section: Urothelial PTEN deletion increases UTI susceptibility "has" should be "have"

Discussion:

Page 9: "in" needs capitalization

Figure Legends:

Figure 4B: I think this is representative "q"PCR not "PCR"

Reviewer #3 (Comments to the Authors (Required)):

This manuscript details the contribution of PTEN to bladder infection by uropathogenic *E. coli* (UPEC) using human bladder urothelial cells as well as a mouse model of UTI. The impact of loss of PTEN was examined using siRNA in bladder cells and via a urothelium-specific PTEN knockout mouse. Through a combination of techniques, the authors determined that loss of PTEN increases UPEC invasion of bladder epithelial cells and results in a ~1 log increase in urine and bladder colonization in mice as well as an increase in bladder histopathology and incidence of kidney colonization. The authors further demonstrate that loss of PTEN increases Akt, NFkB, and FAK activation while decreasing IL-6 and IL-8 cytokine production, suggesting dysregulation of antibacterial responses coupled with an increase in cellular processes that facilitate UPEC invasion. The manuscript is well written, conclusions are largely justified, and limitations of the study are appropriately acknowledged in the discussion. The purpose of the below comments is to further improve the impact and rigor of the manuscript.

1. Overall, the manuscript would be strengthened by inclusion of a model figure that links all of the players identified in the current body of work to other known factors involved in UPEC internalization and defense mechanisms.
2. The authors note on page 10 that UPEC type 1 fimbriae bind beta-1-integrins and stimulate a signaling cascade involving PI3K and FAK to facilitate internalization. This suggests that the enhanced invasion of UPEC when PTEN is lacking and FAK is activated may be driven by type 1 fimbriae. Do the authors think that type 1 fimbriae are required for the increased invasion that occurs when PTEN is knocked down?
3. As FAK regulates cytoskeletal remodeling and FAK expression is altered even in the absence of infection, it would be interesting to determine if cytoskeletal remodeling is already occurring in the PTEN knockdown cells in the absence of infection, or if changes related to increased UPEC invasion only occur following contact with bacteria.
4. The authors note on page 8 that phagocyte cell populations were comparable between control and pTEN knockout bladders. However, the data in Supplemental Figure 5 show a trend towards increased numbers of monocytes and Ly6C+ macrophages in the knockout. It would be helpful to provide the p-values in this figure to convey the strength of this trend.

5. Figure 3A would be improved by showing expression of all three genes in all siRNA groups (for example, include NFkB1 expression of the FAK knockdown in the middle section of the graph).
6. With respect to data presentation and analysis, all error bars in the figures show standard error of the mean rather than standard deviation, which is the more appropriate measure for variance in these types of data. For analysis of fold change data where the control samples are adjusted to 1, a Wilcoxon signed-rank test or one sided t test against a hypothetical value of 1 would be more appropriate than a Student's t test.
7. The parental strain of mouse needs to be specified in the methods and results.
8. It would also be helpful to include a reference or data demonstrated whether treatment with bpV(HOPic) has any impact on UPEC growth or viability.
9. Supplemental Figure 1 would be strengthened by inclusion of densitometry for panel A.
10. Supplemental Figure 6 is not referenced in the main text.
11. There is bolded text at the end of each Supplemental Figure legend that often refers to the wrong figure.

June 8, 2025

RE: **LSA-2025-03292-T**

Dear Dr. Hebbar and the Editorial Board for *Life Science Alliance*,

We are pleased to resubmit our manuscript entitled "*PTEN modulates urinary tract infection susceptibility and shapes urothelial antibacterial defenses*" for your consideration for publication. We appreciate the reviewers' comments and have addressed all their comments in the revised manuscript (notable changes in **BLUE** font), as detailed below in the **POINT-BY-POINT RESPONSES TO THE REVIEWERS**. We believe these modifications have strengthened our manuscript.

Sincerely,

Aaron Simoni and John David Spencer on behalf of the other authors

POINT-BY-POINT RESPONSES TO THE REVIEWERS:

Reviewer #1

Comment 1: Page 5: "These results indicate that PTEN loss or impaired activity enhances UPEC susceptibility." I suggest changing to "enhances susceptibility of the urothelium to UPEC invasion" for clarity.

Response: We have made this change.

Comment 2: Page 5: "STRING protein-protein interaction and signaling enrichment analyses of these significantly differentially expressed genes revealed a dysregulated network with Akt and NFkB as central activated nodes in PTEN deficient cells." I do not see NFkB as one of the nodes in the figure, as is suggested should be true by the text. Please specify which nodes implicate NFkB and why.

Response: We have amended the result section to include this clarification on page 5.

Comment 3: Page 7: I see errors in the description of Fig 3B. For example, it is stated that "PTEN and FAK silencing suppressed UPEC invasion comparable to FAK silencing alone" but this comparison is not indicated by a bar or star(s) on the graph. I think the authors likely meant to say comparable to PTEN silencing alone. Also, it is stated that PTEN and NFkB silencing "remained elevated compared to controls" but the PTEN + NFkB and NTC bars are roughly the same height and no significant difference is indicated on the graph. Please make the necessary corrections to the graph and/or text descriptions.

Response: Thank you for highlighting these inconsistencies. We have revised the Results section on page 7 to accurately reflect the data shown in **Figure 3**. Specifically, we corrected the statement to indicate that "PTEN and FAK silencing suppressed UPEC invasion comparable to PTEN silencing alone," as supported by the statistical comparisons presented. Additionally, we removed the description stating that PTEN and NFkB silencing "remained elevated compared to controls," as this was not supported by a statistically significant difference on the graph. These changes ensure alignment between the text and the figure.

Comment 4: Page 9: "Deficient PTEN expression also augments FAK - an established mediator of UPEC invasion into the urothelium." It would be helpful for the authors to provide a few more sentences describing prior studies that established FAK as a mediator of UPEC invasion. What experiments were done, etc. This will allow the reader to place this study into the broader context of the field.

Response: We have expanded the discussion on page 11 to include this information.

Comment 5: Page 9: "PTEN deletion enhances Salmonella typhimurium susceptibility." And "PTEN loss exhibit increased susceptibility to Mycoplasma spp. and Mycobacterium bovis Bacillus Calmette-Guerin infection." Were these intracellular infections, like with UPEC in the present study? Again, a few more details of the experiments would be helpful.

Response: We have added additional experimental details to this section of the Discussion on page 9.

Comment 6: Page 10: "The plant-derived phenolics caffeic acid phenethyl ester (CAPE) and resveratrol inhibit FAK and reduce UPEC invasion of human bladder epithelial cells.50 Additionally, Lactuca indica extract from Vietnamese dandelion interferes with bacterial adherence and FAK activation, thereby reducing UPEC invasion." Since the authors have mentioned these compounds, it begs the question whether they would work to reduce UPEC invasion in vivo. While I do not think it is a requirement for this manuscript to have additional data, it would certainly be nice if a FAK inhibitor reversed the UTI severity phenotype in the PTEN conditional knockout mice. I think many readers will have this same thought. I encourage the authors comment on whether they have attempted this in their mice, why they have not chosen to, or whether they plan to in future studies.

Response: We have expanded our discussion to include this information on page 12.

Comment 7: Finally, kudos to the authors (or the journal if it was in the author instructions?) for providing the helpful information in the supplemental figure legends indicating which main figures they connect with. This is very helpful, and I wish all supplemental figure legends required this.

Response: Thank you, we are pleased to learn this was helpful!

Reviewer #2

Major Issues/Key Experiments:

Comment 1: Additional details for figure 2 about controls used would help clarify the results for the reader. Specifically, for the inflammatory genes differentially regulated (2A and B) between NTC and PTEN UPEC infected cells, are any of these genes differentially regulated in the uninfected cells prior to infection.

Response: We appreciate the reviewer's interest in understanding baseline gene expression prior to infection. While we did not profile transcript expression of inflammatory genes in non-infected NTC and PTEN deficient cells, we instead chose to validate and extend our findings by assessing protein expression of key inflammatory mediators identified in **Figure 2A/B**. As shown in **Figures 2C/D**, we evaluated expression of these proteins in both non-infected and infected

cells. We believe the protein-level data provide stronger functional relevance than transcript alone and help confirm differential regulation in the presence and absence of infection.

Comment 2: Small comment, but 2B only displays NFkB-mediated cytokines not NFkB itself. The text could be clarified in the sentence prior to 2B denoted in the manuscript (page 5) and/or that designation should be added to the figure legend to clarify which are NFkB related genes.

Response: Thank you for this comment as this was also noted by Reviewer #1. We have amended the results section on page 5 to include this information.

Comment 3: It would strengthen the conclusions that the pathway is linear from PTEN to NFkB/FAK if the authors included the relative transcript of PTEN during the silencing of downstream targets, confirming no feedback/crosstalk (figure 3). Same for NFkB and PTK2 silencing (page 6).

Response: Thank you for this helpful suggestion. Reviewer #3 had a similar comment. To evaluate potential feedback or crosstalk between PTEN, NFkB, and FAK, we now include new data in **Supplemental Figure 3** showing transcript levels of PTEN during NFkB and FAK silencing, as well as NFkB and PTK2 during PTEN silencing. These data confirm that silencing of individual or combined targets does not alter the expression of upstream or parallel pathway members, supporting a linear signaling relationship.

Comment 3: The authors discuss the implications for altered IL-8 and its receptor CXCR1 in human UTI and specifically, the link to progression to pyelonephritis. Additionally, their human cell experiments show IL-8 is altered, while their mouse experiments indicate CXCI-1 is altered. While UPEC dissemination beyond the urinary tract in these models is rare (<5%), assessing dissemination in PTEN KO mice (spleen/heart/liver) could provide additional insight into mechanisms essential for preventing the spread from the bladder, which is arguably the most serious complication of UTI.

Response: We agree that bacterial dissemination is indeed a critical clinical complication, while it remains rare in murine models of UTI on the C57BL/6 background. In our current study, we focused on bladder and kidney tissues, as well as urine, to define the impact of PTEN loss on UPEC susceptibility and local host responses. Unfortunately, we did not assess UPEC burden in secondary organs such as the liver, spleen, or heart. While we recognize that evaluating dissemination in PTEN-deficient mice could yield valuable insight into systemic host defense, the time and cost required to regenerate sufficient mouse cohorts and perform repeat infections precluded this analysis. We have now acknowledged this limitation in the revised Discussion.

Minor Comments:

Page 8, Section: Urothelial PTEN deletion increases UTI susceptibility "has" should be "have"

Response: This has been corrected.

Page 9: "in" needs capitalization

Response: This has been corrected.

Figure 4B: I think this is representative "q"PCR not "PCR"

Response: Figure 4B shows conventional PCR, not qPCR (or qRT-PCR). We have amended the figure legend.

Reviewer #3

Comment 1: Overall, the manuscript would be strengthened by inclusion of a model figure that links all of the players identified in the current body of work to other known factors involved in UPEC internalization and defense mechanisms.

Response: This has been added as **Supplemental Figure 8**.

Comment 2: The authors note on page 10 that UPEC type 1 fimbriae bind beta-1-integrins and stimulate a signaling cascade involving PI3K and FAK to facilitate internalization. This suggests that the enhanced invasion of UPEC when PTEN is lacking and FAK is activated may be driven by type 1 fimbriae. Do the authors think that type 1 fimbriae are required for the increased invasion that occurs when PTEN is knocked down?

Response: We have added language on page 11 of the revised discussion to address this comment.

Comment 3: As FAK regulates cytoskeletal remodeling and FAK expression is altered even in the absence of infection, it would be interesting to determine if cytoskeletal remodeling is already occurring in the PTEN knockdown cells in the absence of infection, or if changes related to increased UPEC invasion only occur following contact with bacteria.

Response: To address whether cytoskeletal remodeling occurs in PTEN deficient cells in non-infected cells, we evaluated the expression of uroplakin, cell junction, and focal adhesion genes. This analysis was shown in Supplemental Figure 2 of our initial submission. In this revision, we expanded our evaluation and identified differential expression of cytoskeletal adaptor genes (*TLN1* and *TNS1*). These data, now included in **Supplemental Figure 2** and highlighted in the results section on page 6, support early remodeling events in non-infected PTEN deficient cells. While we did not perform actin staining or focal adhesion imaging in this study, future work could examine whether these transcriptional changes correspond to phenotypic alterations in cytoskeletal architecture.

Comment 4: The authors note on page 8 that phagocyte cell populations were comparable between control and pTEN knockout bladders. However, the data in Supplemental Figure 5 show a trend towards increased numbers of monocytes and Ly6C+ macrophages in the knockout. It would be helpful to provide the p-values in this figure to convey the strength of this trend.

Response: In the resubmission, we have added the calculated *P*-values to **Supplemental Figure 6** to convey the statistical significance of trends in phagocyte populations.

Comment 5: Figure 3A would be improved by showing expression of all three genes in all siRNA groups (for example, include NFKB1 expression of the FAK knockdown in the middle section of the graph).

Response: We appreciate the recommendation from yourself and reviewer #2 to include gene expression data for all three targets across all siRNA groups. In response, we have added

Supplemental Figure 3, which shows qRT-PCR data for PTEN, NFkB, and PTK2 expression in each knockdown condition. This confirms successful silencing and the specificity of each siRNA pool, strengthening the conclusions drawn from **Figure 3A**.

Comment 6: With respect to data presentation and analysis, all error bars in the figures show standard error of the mean rather than standard deviation, which is the more appropriate measure for variance in these types of data. For analysis of fold change data where the control samples are adjusted to 1, a Wilcoxon signed-rank test or one sided t test against a hypothetical value of 1 would be more appropriate than a Student's t test.

Response: We thank the reviewer for this suggestion. In this study, our experiments included both *in vitro* tissue culture models and mouse biological replicates. Our primary interest was to assess differences between group means, rather than to characterize sample variability. We therefore chose to display data as mean \pm SEM, as SEM more accurately represents the uncertainty in the estimate of the group mean, particularly when the goal is to infer differences between conditions. In contrast, SD is more appropriate when the objective is to show the dispersion or variability among individual data points. To enhance transparency, we included individual data points in all graphs, enabling readers to visually assess both the range and distribution of the data.

To ensure our approach was appropriate, we also reviewed our statistical methods with two local biostatisticians, both of whom confirmed that mean \pm SEM is most appropriate given the study design and analytical goals. This is supported in the literature (PMID: 34012702), where the authors conclude: "From a biostatistics point of view, we favor the use of SEM over that of SD, for describing scientific results under most circumstances."

We also appreciate the thoughtful comment regarding statistical analysis of our fold change data. We assume this is regarding the qRT-PCR data and/or the densitometry data. While many of the values appear close to 1, we would like to clarify that they are not mathematically constrained or normalized to a fixed value. Rather, these values are derived from independent biological replicates – either from densitometry normalized to GAPDH (Western blotting) or from qRT-PCR analyzed using the $2^{-\Delta\Delta Ct}$ method. In both cases, control values vary between replicates and are not artificially set to 1.

Because the resulting values are continuous and biologically derived, we assessed data distribution using the Shapiro-Wilk normality test. For datasets that met parametric assumptions, we used unpaired Student's t-tests. In cases where normality was not met or data were not suitable for parametric analysis, we applied non-parametric tests such as the Mann-Whitney *U* test, which is appropriate for comparing two independent groups. We did not use the Wilcoxon signed-rank test, as our comparisons were not paired or matched to a hypothetical value.

We have clarified the statistical tests used in the figure legends. We have also ensured that the statistical methods used are appropriate for the nature and distribution of each dataset.

Comment 7: The parental strain of mouse needs to be specified in the methods and results.

Response: This has been added.

Comment 8: It would also be helpful to include a reference or data demonstrated whether treatment with bpV(HOpic) has any impact on UPEC growth or viability.

Response: This (new) data has been added to **Supplemental Figure 1**.

Comment 9: Supplemental Figure 1 would be strengthened by inclusion of densitometry for panel A.

Response: Densitometry has been added to **Supplemental Figure 1A**. As shown, bpV(HOpic) increases pAKT levels but does not affect total PTEN expression. bpV(HOpic) is a cell-permeable, vanadium-based inhibitor that suppresses PTEN phosphatase activity by reversibly oxidizing its catalytic cysteine residue. This post-translational mechanism does not alter PTEN transcription or protein stability, and thus total PTEN expression is not expected to change with treatment.

Comment 10: Supplemental Figure 6 is not referenced in the main text.

Response: Supplemental Figure 6 was referenced in the methods section of the initial submission, and we have now referenced it in the results section as well as the methods section of the revision. Due to the generation of new data, this supplemental data is now shown in **Supplemental Figure 7**.

Comment 11: There is bolded text at the end of each Supplemental Figure legend that often refers to the wrong figure.

Response: We have rechecked the bolded text at the end of the Supplemental Figure legends to ensure it references back to the appropriate data presented main figures. We agree with Reviewer #1 that this text can help guide the reader.

July 3, 2025

RE: Life Science Alliance Manuscript #LSA-2025-03292-TR

Dr. John David Spencer
Nationwide Children's Hospital
Pediatrics
700 Children's Drive
Columbus, OH 43016

Dear Dr. Spencer,

Thank you for submitting your revised manuscript entitled "PTEN modulates urinary tract infection susceptibility and shapes urothelial antibacterial defenses". The revised manuscript was re-evaluated by the three original reviewers. In line with their comments, we would be happy to publish your paper in Life Science Alliance pending final revisions necessary to meet our formatting guidelines.

- Please add call-out for Figures S1A-D; S2A-B; S4A-B; S5A-B and S7A-F to your main manuscript text.
- We encourage you to revise the figure legend for figure S1 such that the figure panels are introduced in alphabetical order.
- Please add molecular weight markers for all Western Blot images in the figures.
- Please include a scale bar for all microscopic images, and define-scale bars in the associated legend/figure (Figures 4, 5, S4, and S7).
- In the methods section, please specify the objective (with NA) used for microscopy. Please provide details for histology (related to stains used)
- Please include a 'Data Availability' section, specifying if you are willing to provide source data or have done so in supplementary information.
- Please attend to grammatical errors indicated by Reviewer 2.
- "The paper explained" section is not a requirement at LSA and can be removed.
- We leave it to your decision if you wish to use Fig. S8 as a Graphical Abstract (GA). Should you choose to have it as a GA instead, you must completely remove Fig. S8, associated legend, and callout, from the main manuscript. Please also upload it with the designation 'Graphical Abstract'.
- Please add a Category for your manuscript in our system.
- Please be sure that the authorship listing and order are correct and match between the system and the manuscript file.

A. FINAL FILES:

- An editable version of the final text (.DOC or .DOCX) is needed for copyediting (no PDFs).
- High-resolution figure, supplementary figure and video files uploaded as individual files: See our detailed guidelines for preparing your production-ready images, <https://www.life-science-alliance.org/authors>
- Summary blurb (enter in submission system): A short text summarizing in a single sentence the study (max. 200 characters)

including spaces). This text is used in conjunction with the titles of papers, hence should be informative and complementary to the title. It should describe the context and significance of the findings for a general readership; it should be written in the present tense and refer to the work in the third person. Author names should not be mentioned.

B. MANUSCRIPT ORGANIZATION AND FORMATTING:

Sincerely,

Sarita Hebbar, PhD
Scientific Editor
Life Science Alliance
<http://www.lsajournal.org>

Reviewer #1 (Comments to the Authors (Required)):

The authors have adequately addressed all of my comments and the comments from the other two reviewers. I have not further concerns. It is a good, solid paper of interest to the field.

Reviewer #2 (Comments to the Authors (Required)):

The authors present a robust study of mechanism by which PTEN and its downstream targets impact susceptibility to UTIs caused by UPEC. This study builds on their previous work dissecting this pathway and demonstrates how PTEN works upstream to regulate PI3K/Akt, and therefore its downstream effectors NFkB and FAK, to modulate UTI susceptibility. This work represents an important step towards understanding the host-pathogen mechanisms that contribute to host susceptibility to UPEC UTI.

The authors have addressed my major concerns.

Only minor grammatical errors remain: line 184 "observed", line 378 "F", line 733 "SEZM" etc.

Reviewer #3 (Comments to the Authors (Required)):

The revision has sufficiently addressed all main points raised by the reviewers.

July 9, 2025

RE: **LSA-2025-03292-TR**

Dear Dr. Hebbar and the Editorial Board for *Life Science Alliance*,

We are pleased to resubmit our manuscript entitled "*PTEN modulates urinary tract infection susceptibility and shapes urothelial antibacterial defenses*" for your consideration for publication. We have addressed the editorial points as outlined below:

Sincerely,

Aaron Simoni and John David Spencer on behalf of the other authors

EDITORIAL POINTS:

Point 1: Please add call-out for Figures S1A-D; S2A-B; S4A-B; S5A-B and S7A-F to your main manuscript text.

Response: We have updated this in the results section.

Point 2: We encourage you to revise the figure legend for figure S1 such that the figure panels are introduced in alphabetical order.

Response: We have revised the legend for Supplemental Figure 1.

Point 3: Please add molecular weight markers for all Western Blot images in the figures.

Response: We have added molecular weight markers for the presented representative Western blots.

Point 4: Please include a scale bar for all microscopic images, and define-scale bars in the associated legend/figure (Figures 4, 5, S4, and S7).

Response: These are included in the submitted figures.

Point 5: In the methods section, please specify the objective (with NA) used for microscopy. Please provide details for histology (related to stains used)

Response: We have added the requested information to the methods section.

Point 6: Please include a 'Data Availability' section, specifying if you are willing to provide source data or have done so in supplementary information.

Response: We have added this section to the revised submission.

Point 7: Please attend to grammatical errors indicated by Reviewer 2.

Response: The three grammatical errors have been corrected.

Point 8: "The paper explained" section is not a requirement at LSA and can be removed.

Response: This section has been removed.

Point 9: We leave it to your decision if you wish to use Fig. S8 as a Graphical Abstract (GA). Should you choose to have it as a GA instead, you must completely remove Fig. S8, associated legend, and callout, from the main manuscript. Please also upload it with the designation 'Graphical Abstract'.

Response: We elected to keep this figure in the main body of the paper since the right panel captures data from prior publications.

Point 10: Please add a Category for your manuscript in our system.

Response: This has been added.

Point 11: Please be sure that the authorship listing and order are correct and match between the system and the manuscript file.

Response: The authorship reflect the manuscript file.

July 15, 2025

RE: Life Science Alliance Manuscript #LSA-2025-03292-TRR

Dr. John David Spencer
Nationwide Children's Hospital
Pediatrics
700 Children's Drive
Columbus, OH 43016

Dear Dr. Spencer,

Thank you for submitting your Research Article entitled "PTEN modulates urinary tract infection susceptibility and shapes urothelial antibacterial defenses". It is a pleasure to let you know that your manuscript is now accepted for publication in Life Science Alliance. Congratulations on this interesting work.

DISTRIBUTION OF MATERIALS:

Again, congratulations on a very nice paper. I hope you found the review process to be constructive and are pleased with how the manuscript was handled editorially. We look forward to future exciting submissions from your lab.

Sincerely,

Sarita Hebbar, PhD
Scientific Editor
Life Science Alliance
<http://www.lsajournal.org>